# Modified W-type configuration for a single-phase reduced parts count 81-level inverter

**Sajjad Ahmed**, **Muhammad Asghar Saqib***, **Syed Abdul Rahman Kashif**

Department of Electrical Engineering, University of Engineering and Technology, Lahore, Punjab, Pakistan

☯ These authors contributed equally to this work.
* saqib@uet.edu.pk

**Data Availability Statement:** All relevant data are within the paper and its Supporting information files.

**Funding:** The authors received no specific funding for this work.

## Abstract

The technology of modern power systems is revolutionizing as renewable energy sources are being integrated with electric power grids. In the form of inverters, power electronic converters are becoming an integral part of power systems due to their massive demand for grid integration of photovoltaic (PV) systems. Existing multilevel inverter topologies either require an output filter to get a sinusoidal voltage or generate a higher number of output voltage levels at the expense of many hardware resources. This paper presents a new single-phase 81-level inverter configuration with the name given as 'Modified W-Type Multilevel Inverter'. The proposed inverter configuration uses only eighteen power switches and four DC voltage sources to generate an 81-level output voltage approaching a sinusoidal waveform without an output filter. The general design equations are developed to calculate the number of switches, the number of voltage levels, and the number of DC sources for the proposed configuration. Loss and efficiency analysis is carried out that verifies a good practical efficiency of the proposed inverter configuration during the dynamic operation. A comparative analysis with the existing MLI topologies is also carried out that validates the effectiveness and novelty in reducing parts count and higher number of voltage levels. The proposed topology offers 1.04% total harmonic distortion of the output voltage which is within the benchmarks specified for grid integration without any filter requirements. The proposed inverter configuration is simulated in MATLAB/Simulink, and the results are validated by the design and development of a hardware prototype.

## 1 Introduction

The integration of renewable energy sources with electric power grids is one of the foundations for the future transformation of power systems. The conventional grids are being converted into smart grids with the help of modern technology. The world is also concerned about the increasing environmental pollution and the energy demand due to a boom in the population and developments. To overcome challenging energy demands and the problems of environmental pollution, green energy solutions such as renewable energy resources are being integrated with electric power grids. This integration requires an optimized power

**Competing interests:** The authors have declared that no competing interests exist.

**Abbreviations:** $a(t)$, $b(t)$, ON-state number of switches, body diodes at a given time; $N_D$, $N_{GD}$, $F_{CPL}$, Number of diodes, number of gate drivers, components per level factor; $N_L$, $N_{DC}$, $N_S$, Number of levls, number of DC sources, number of switches; $N_{on}$, $N_{off}$, Number of switches turning-ON and OFF respectively.; $P_{c,sw}$, $P_{c,d}$, Conduction loss of switch, conduction loss of body diode; $P_{loss}$, $THD$, $\eta$, Total power loss, total harmonic distortion, efficiency; $P_{on,sw}$, $P_{off,sw}$, Switching loss during turn-ON, switching loss during turn-OFF; $R_{on,sw}$, $R_{on,d}$, ON-resistance of switch, ON-resistance of body diode; $t_{on}$, $t_{off}$, $T$, Turn-ON time, turn-OFF time, time period; $TSV$, $V_{sw}$, Total standing voltage, average voltage of switch during switching; $V_i$, $V$, Blocking voltage of switch 'i', lowest DC voltage in the circuit; $V_o$, $I$, $\beta$, Output voltage, switch current, switch constant; $V_{on,sw}$, $V_{on,d}$, ON-state voltage of switch, ON-state voltage of body diode.

flow and satisfactory compliance with benchmarks. The power inverters are extensively used in the integration of renewable sources (such as solar photovoltaic (PV) systems, variable speed wind turbine systems, and fuel cells), HVDC power transmission systems, power system compensators, industrial machine drives, Flexible AC Transmission Systems (FACTS), and nowadays mostly in hybrid electric vehicles [1]. To meet the voltage quality benchmarks and reduce the total harmonic distortion (THD), multilevel inverters (MLIs) are extensively deployed for the grid integration of renewable energy sources. MLIs offer several advantages over conventional H-bridge two-level inverters, such as small filter requirement, low dv/dt stress across power switches (MOSFETs/IGBTs), low power rating of switches, and low electromagnetic interference (EMI). The switching losses in MLIs are also more minor than those in conventional two-level inverters because of the effective low switching frequency of devices. Recent MLIs have been developed using the variation in the structure of the conventional MLIs such as Flying Capacitor (FC), Neutral Point Clamped (NPC), and Cascaded H-bridge (CHB) topologies. MLIs have gained central importance in the last few years as the demand for green energy generation-based medium power systems is increasing every day [2]. However, apart from the merits of MLIs, they have an increased number of switch counts, isolated DC sources, and isolated gate drivers. There are several latest MLI topologies available in the literature [3–6]. In some MLIs, a primary unit cell is repeated (modular) to obtain more number of levels, and a higher output voltage [7, 8] whereas some other topologies (non-modular) are dedicated to providing a fixed number of voltage levels depending on the structure of the MLIs [9–11].

Based on the structure of MLIs, their configuration can be classified as either symmetric (having an equal value of DC voltage sources) or asymmetric (having an unequal value of DC voltage sources). In addition to the classification related to symmetry, there is another classification of MLIs based upon the need for an H-bridge to obtain the negative levels as some topologies cannot generate both positive and negative levels without using an H-bridge. In [12], a cascaded MLI using the modular approach based upon its basic unit is presented. The basic unit contains nine switches and two isolated DC sources to produce nine output voltage levels. Using the cascade connection of two basic units, seventeen levels are achieved and the switch count rises to sixteen with four isolated DC voltage sources being in operation. In [13], an envelope-type module is introduced as a basic building block, which is then used to develop an MLI with a cascaded connection of the selected modules. As this topology generates all possible levels, there is no need for an H-bridge, and hence the power switches have reduced voltage stress. The magnitude of isolated DC voltage sources is also unequal, and hence it can be classified as an asymmetric configuration.

In [14], a primary unit cell consisting of two DC voltage sources, a bidirectional switch, and two power switches has been proposed. To increase the number of voltage levels, the basic modules in this configuration are arranged in a cascaded connection without an H-bridge. This configuration requires a lesser number of gate drive circuits as compared to that in the CHB configuration. A new topology with Square T-Type configuration was presented in [15]. It uses four DC power sources with twelve switches to generate seventeen levels at the output. Another modified topology of MLI in [15] named as K-type is presented in [16]. It uses two DC power sources, two capacitors, and fourteen power switches to generate thirteen output voltage levels. In this configuration, the two power sources are replaced with two capacitors.

In [17], a T-type topology without the use of an H-Bridge was presented to generate different higher levels with ten power switches at various loading conditions. A 31-level asymmetric MLI topology is presented in [18] that uses ten switches with four power supplies to generate a

31-level output. The main drawback of this configuration is that it needs nine isolated gate drive circuits for proper operation. In [19], an MLI topology that is appropriate for both high and low switching frequencies has been presented. This topology uses eight switches, three DC power sources, and seven gate drivers to generate 11-level output. Besides all of the above-discussed topologies, there are some other MLI topologies presented in [4, 20–26] that have a reduced number of device counts and low voltage stress across the power switches.

All the topologies discussed above have the main disadvantage that as the number of levels is increased, the count for power switches shoots up, and the overall cost and size of the inverter increase critically. Increasing the number of power switches to obtain more levels further increases the number of isolated gate drivers, the number of DC voltage sources, and the software complexity to control the power switches.

This paper proposes a new Modified W-type Multilevel Inverter (MWMLI) topology with reduced parts count. The proposed topology uses eighteen unidirectional switches, four DC power sources, and eight isolated gate drive circuits to produce 81-levels of voltage at the output with the help of an H-bridge. The proposed inverter utilizes significantly less number of switches, number of gate drive circuits, and DC voltage sources to generate the output voltage approaching a sinusoidal waveform with a low THD. The reduction in the hardware components count further reduces the software complexity for pulse width modulation (PWM) generation. The general design equations for the proposed topology are developed along with the equations for loss and efficiency calculation. Comparative analysis is also carried out to compare several important parameters of the proposed topology, such as number of switches, number of DC sources, components per level factor, number of diodes, number of capacitors, THD, and total standing voltage (TSV) with the existing topologies.

The paper is organized as follows. Section 2 discusses the proposed 81-level MLI topology in detail and the switching states for all the possible voltage levels. The loss and efficiency analysis of the proposed MWMLI is carried out in section 3. Section 4 discusses the comparative analysis and compares various features of the proposed MWMLI with existing MLI topologies. Section 5 discusses the results obtained using simulation and hardware prototype along with achievements and drawbacks of the proposed MWMLI topology. Finally, section 6 summarizes the results and confirms the novelty of the proposed inverter topology using the prototype-based experiments.

## 2 Proposed 81-level MWMLI topology

The MWMLI configuration is shown in Fig 1. The name modified W-type suggests that the proposed configuration is somehow similar to the English alphabet 'W'. The proposed configuration uses fourteen switches and four DC power sources, and an H-bridge comprised of four switches at the output of the inverter to generate an 81-level output voltage. The proposed topology can generate positive levels which are then converted to the corresponding negative part using an H-bridge. The proposed topology uses only unidirectional switches with an anti-parallel body diode. Based on the magnitude of DC voltage sources, MLIs are divided into two groups: symmetrical and asymmetrical. The main disadvantage of using symmetrical topology is that the number of power switches also increases as the number of levels is increased. To overcome this issue, the four DC voltage sources have unequal voltages in the proposed topology. The ratio of the voltage for the DC voltage sources $V_4 : V_3 : V_2 : V_1$ is 27:9:3:1. Therefore, 40 voltage levels with a zero level can be obtained by simply adding and subtracting the voltages of four DC sources. The corresponding 40 levels on the negative side of the output voltage are obtained with the help of the H-bridge.

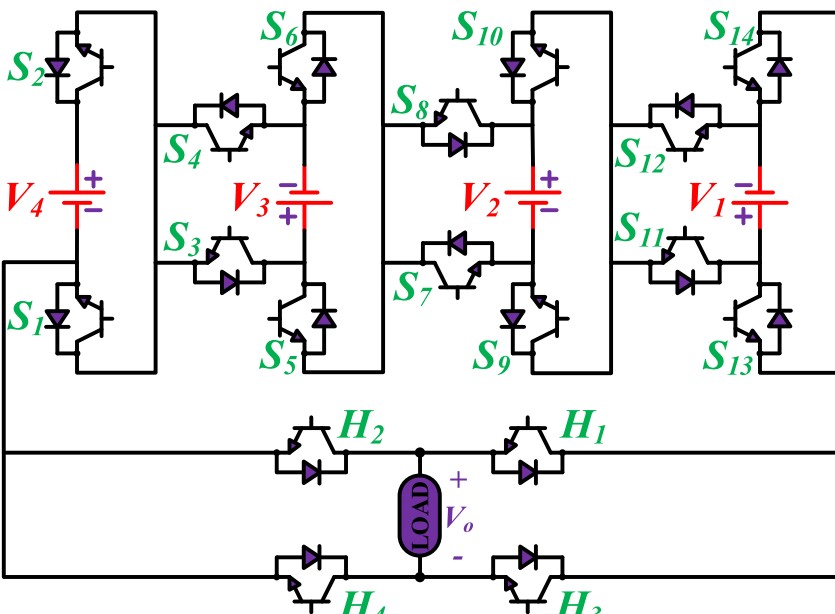

**Fig 1. Proposed configuration of the 81-level MLI.**

The initial zero level is generated by turning ON the switches $S_1$, $S_3$, $S_5$, $S_9$, $S_{11}$, and $S_{13}$. The zero voltage level can also be obtained by turning ON either $H_1$ and $H_3$ or $H_2$ and $H_4$. The switching states for various selected output voltage levels are shown in Figs 2–13. The switching states only show the addition or subtraction of DC power sources. The corresponding positive or negative levels can be obtained by configuring the respective H-bridge switches. The

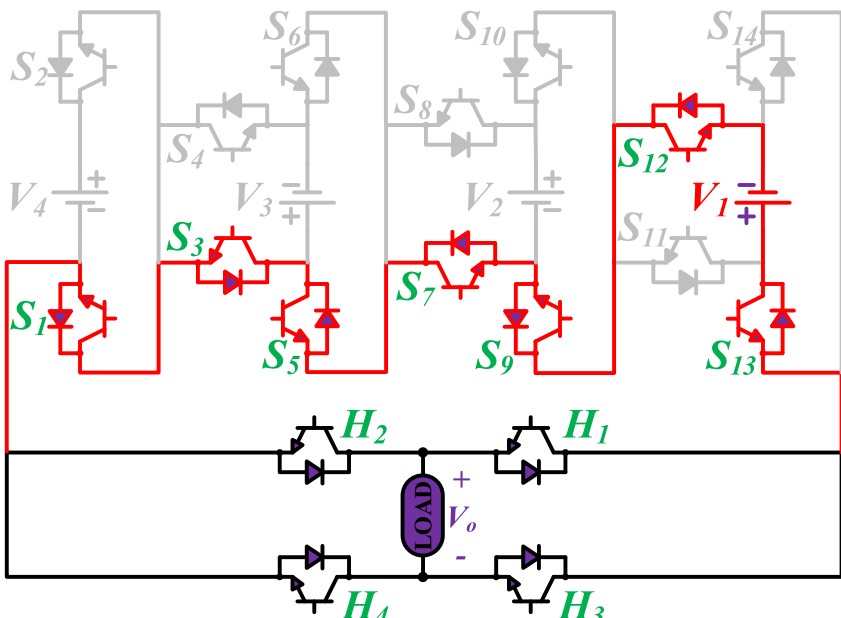

**Fig 2. Switching states for voltage magnitude level 1.**

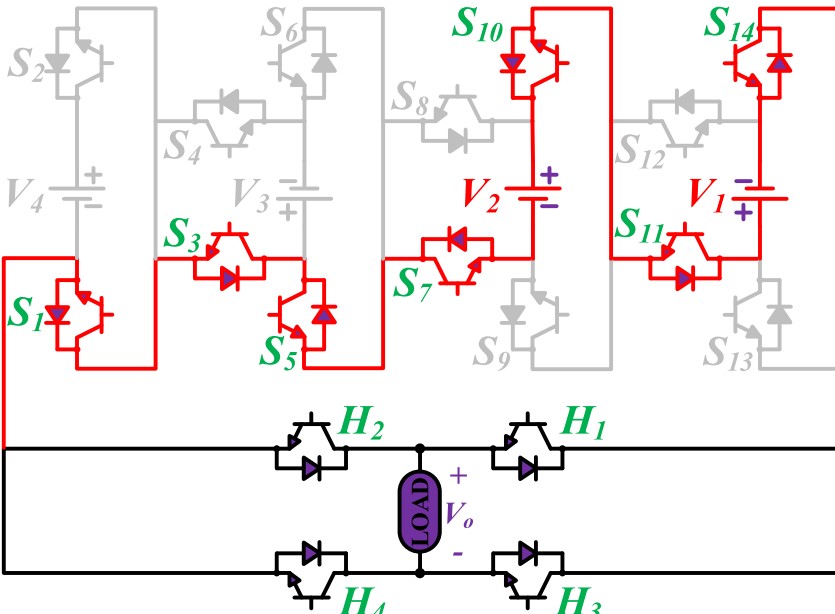

**Fig 3. Switching states for voltage magnitude level 2.**

switching states for all the 81-levels for oddly labeled switches are presented in Table 1. The even labeled switches have opposite states to their respective odd labeled switches. The even labeled switches are exactly above the respective odd labeled switches in Fig 1. The proposed topology of 81-level MWMLI is simulated in MATLAB/Simulink by choosing the value of DC

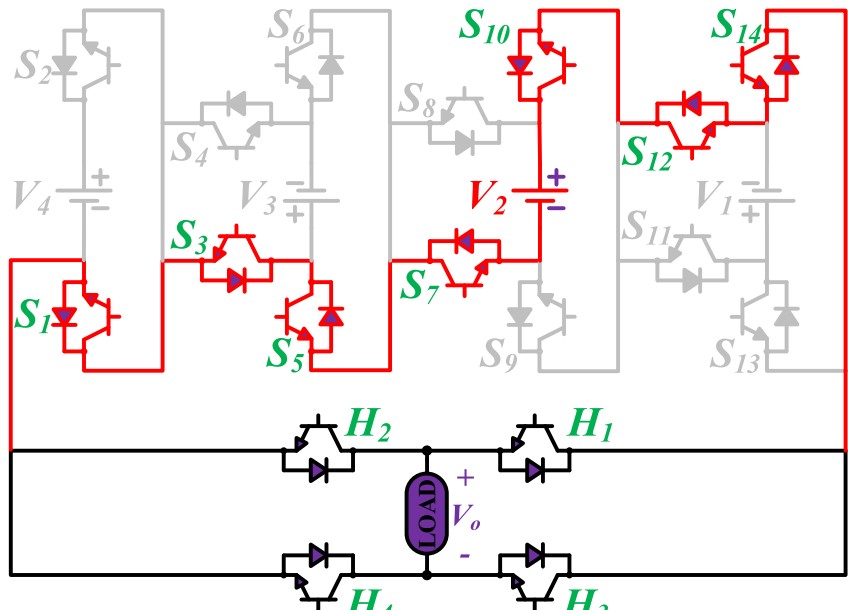

**Fig 4. Switching states for voltage magnitude level 3.**

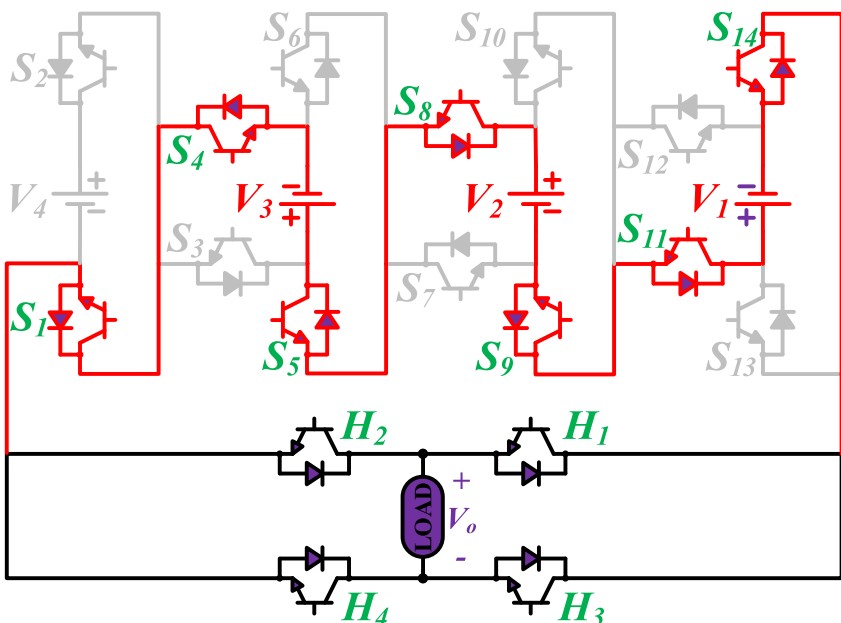

**Fig 5. Switching states for voltage magnitude level 5.**

voltage sources $V_4$:$V_3$:$V_2$:$V_1$ as 270 V:90 V:30 V:10 V. The output voltage waveform with zoomed-in levels is shown in Fig 14. The PWM and the output voltage waveform for resistive, inductive, capacitive, and dynamic load are shown in Figs 15–19 respectively. The harmonic analysis is carried out, and the THD of the output voltage for 81-level MWMLI is computed and is shown in Fig 20. The expressions for the number of DC sources $N_{DC}$ and number of power switches $N_S$ in terms of the number of voltage levels $N_L$ for the proposed topology are

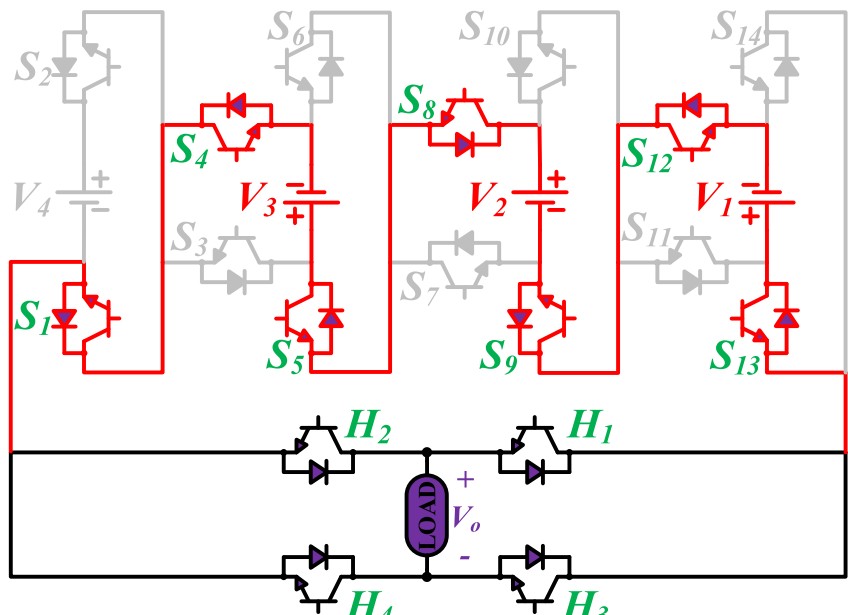

**Fig 6. Switching states for voltage magnitude level 7.**

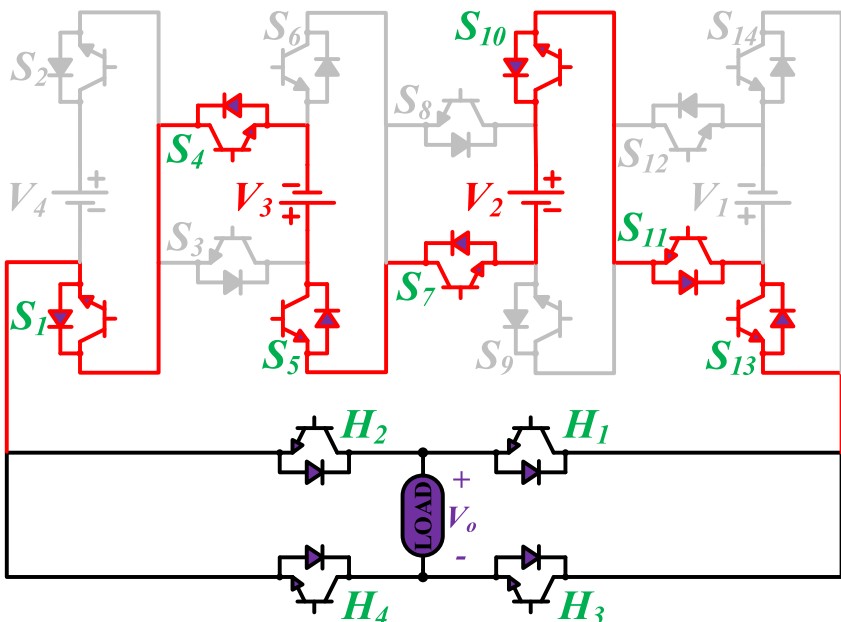

**Fig 7. Switching states for voltage magnitude level 12.**

developed and given as:

$$N_{DC} = \frac{log_{10}(N_L)}{log_{10}(3)} \tag{1}$$

$$N_S = 4\frac{log_{10}(N_L)}{log_{10}(3)} + 2 \tag{2}$$

Conversely, the relationship to find the number of levels in terms of number of DC sources is given as:

$$N_L = 3^{N_{DC}} \tag{3}$$

The magnitude of the output voltage at a specific level is given by the level number multiplied by the lowest DC voltage in the circuit is:

$$|V_o| = Level\ Number \times V_1 \tag{4}$$

where level number is an integer that varies from -40 to 40 for the case of 81-level MWMLI. Therefore, the highest output voltage level has a magnitude that can be calculated as:

$$|V_o(max)| = \frac{(N_L - 1)}{2} \times V_1 \tag{5}$$

TSV is an important parameter for MLIs that is defined as the sum of the maximum blocking voltages across all the power switches in a given inverter topology. It helps select suitable switches to be used in the inverter circuit. TSV also indicates the power losses in the inverter.

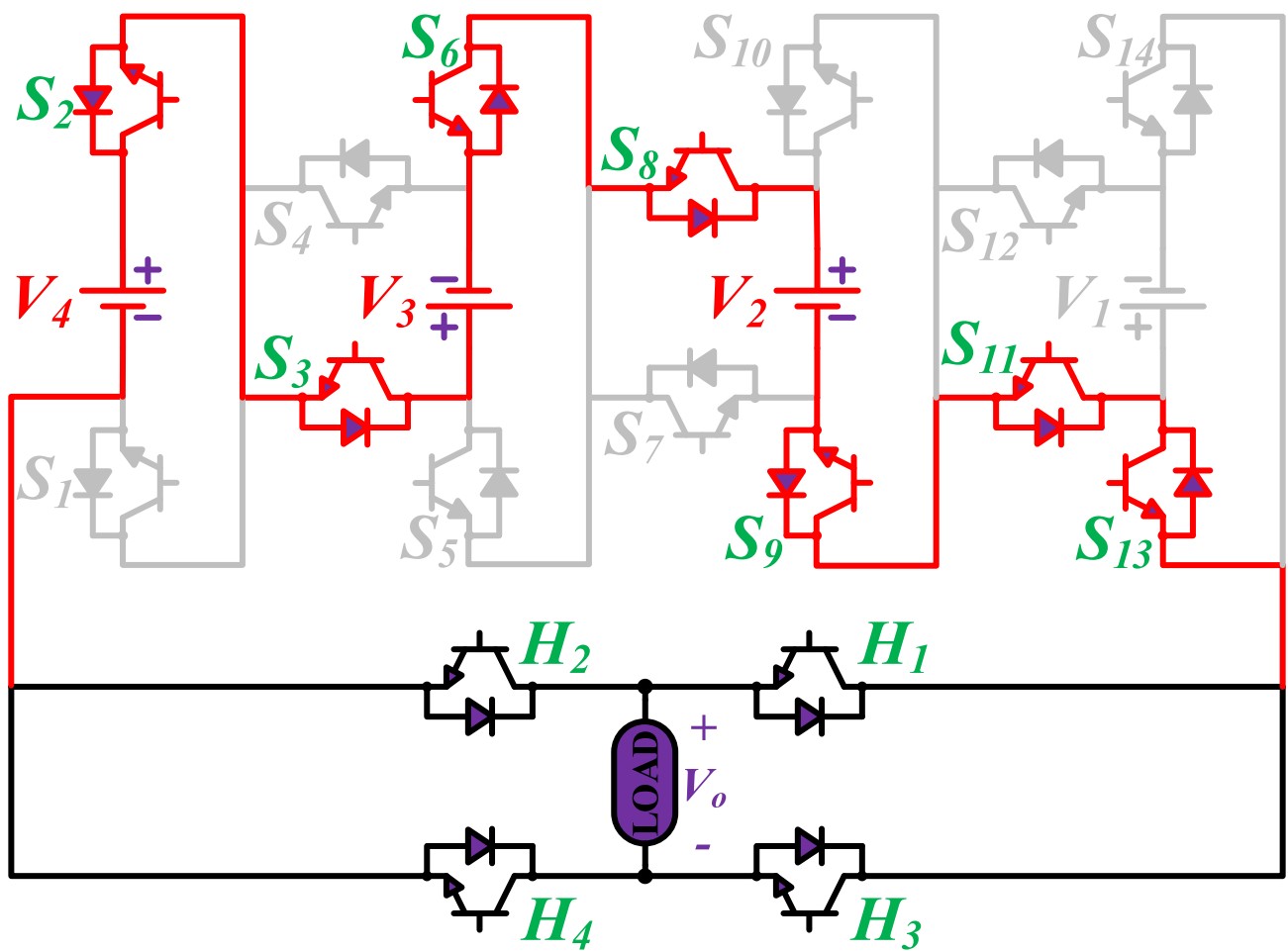

**Fig 8. Switching states for voltage magnitude level 15.**

The TSV for the proposed inverter is calculated as:

$$TSV = 2(V_{S1} + V_{S3} + V_{S5} + V_{S7} + V_{S9} + V_{S11} + V_{S13}) + 4V_{H1} \qquad (6)$$

where, the blocking voltage for all the swithces in the H-bridge is same and is given as:

$$V_{H1} = V_{H2} = V_{H3} = V_{H4} \qquad (7)$$

Using Eq (6), the TSV for the MWMLI is calculated as:

$$TSV = 2(27V + 9V + 9V + 3V + 3V + V + V) + 4(40V) \qquad (8)$$

$$TSV = 266\,V \qquad (9)$$

where $V$ is the lowest DC voltage in the circuit (i.e. $V_1$).

The prototype hardware of the proposed 81-level MWMLI was developed using the components and specifications presented in Table 2. The developed hardware prototype for the experimental verification is shown in Fig 21. The waveforms (for voltage and current) at the output for a resistive and non-linear inductive load are shown in Figs 22 and 23. The waveforms (for voltage and current) at the output for a modulation index (MI) of 0.88 with unity

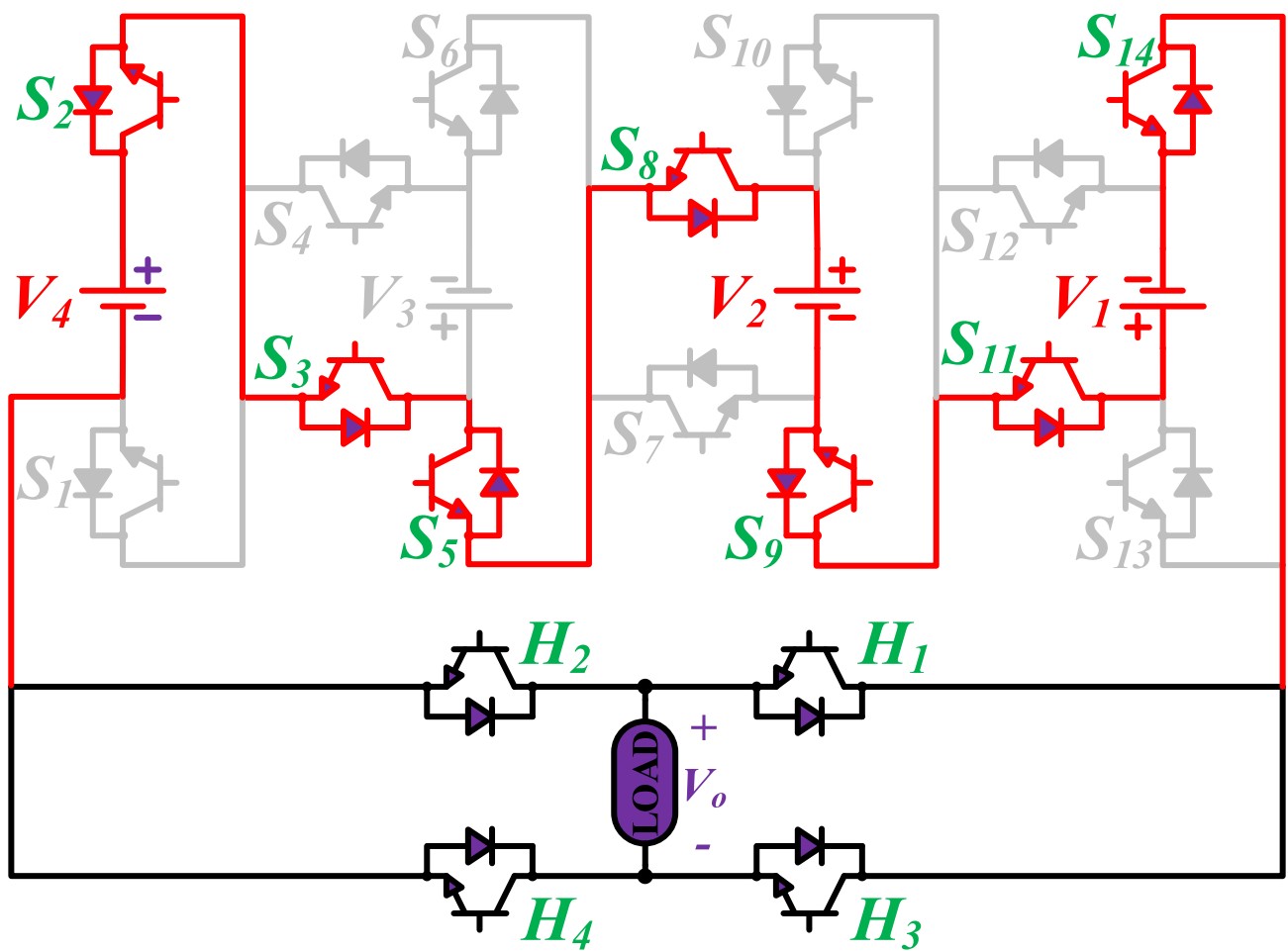

**Fig 9. Switching states for voltage magnitude level 23.**

power factor (PF) are shown in Fig 24 whereas for a modulation index of 0.85 with 0.9 PF lagging are shown in Fig 25.

## 3 Loss and efficiency analysis

The proposed MWMLI can be analyzed for loss and efficiency. There are two types of losses present in the inverter: conduction and switching losses. The conduction losses occur when a switch or its body diode is in conducting state. The conduction loss in a switch and its body diode can be modeled by the following equations respectively:

$$P_{c,sw} = a(t)\left[\frac{1}{2\pi}\int_0^{2\pi}(V_{on,sw}I + R_{on,sw}I^{\beta+1})d(\omega t)\right] \tag{10}$$

$$P_{c,d} = b(t)\left[\frac{1}{2\pi}\int_0^{2\pi}(V_{on,d}I + R_{on,d}I^2)d(\omega t)\right] \tag{11}$$

In Eqs (10) and (11), $V_{on,sw}$ and $V_{on,d}$ are the voltages of the switch and the body diode during ON-state respectively. $R_{on,sw}$ and $R_{on,d}$ are the ON-state resistance of the switch and body

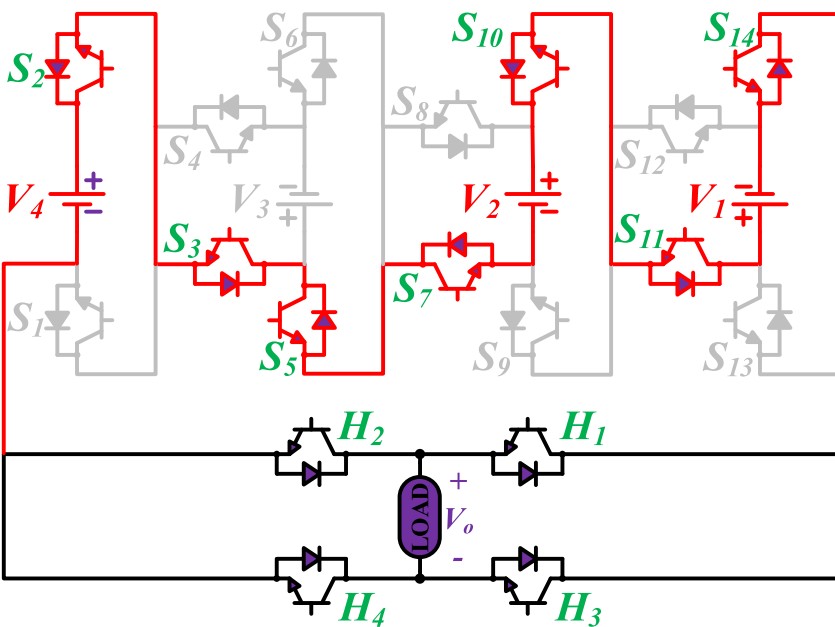

**Fig 10. Switching states for voltage magnitude level 29.**

diode respectively. $a(t)$ and $b(t)$ are the number of switches and number of body diodes respectively that are in ON-state for a given path of current. The current conducted by the switch or diode is assumed to be sinusoidal and $\beta$ is a constant that depends on the properties of the switch. The switching losses during turn-ON ($P_{on,sw}$) time and turn-OFF ($P_{off,sw}$) time are calculated as:

$$
\begin{aligned}
P_{on,sw} &= \frac{N_{on}}{T}\int_0^{t_{on}} V(t).I(t)d(t) \\
&= \frac{V_{sw}.I.t_{on}.N_{on}}{6T}
\end{aligned}
\tag{12}
$$

$$
\begin{aligned}
P_{off,sw} &= \frac{N_{off}}{T}\int_0^{t_{off}} V(t).I(t)d(t) \\
&= \frac{V_{sw}.I.t_{off}.N_{off}}{6T}
\end{aligned}
\tag{13}
$$

where $N_{on}$ and $N_{off}$ are the number of switches that are turning ON and turning OFF respectively during a cycle (with period T). In addition, the turn ON and turn OFF times are represented by $t_{on}$ and $t_{off}$ respectively. Therefore, using Eqs (10)–(13), the total power loss in the proposed MWMLI is given as:

$$
P_{loss} = P_{c,sw} + P_{c,d} + P_{on,sw} + P_{off,sw}
\tag{14}
$$

The efficiency is then calculated as:

$$
\eta = \frac{P_o}{P_o + P_{loss}} \times 100
\tag{15}
$$

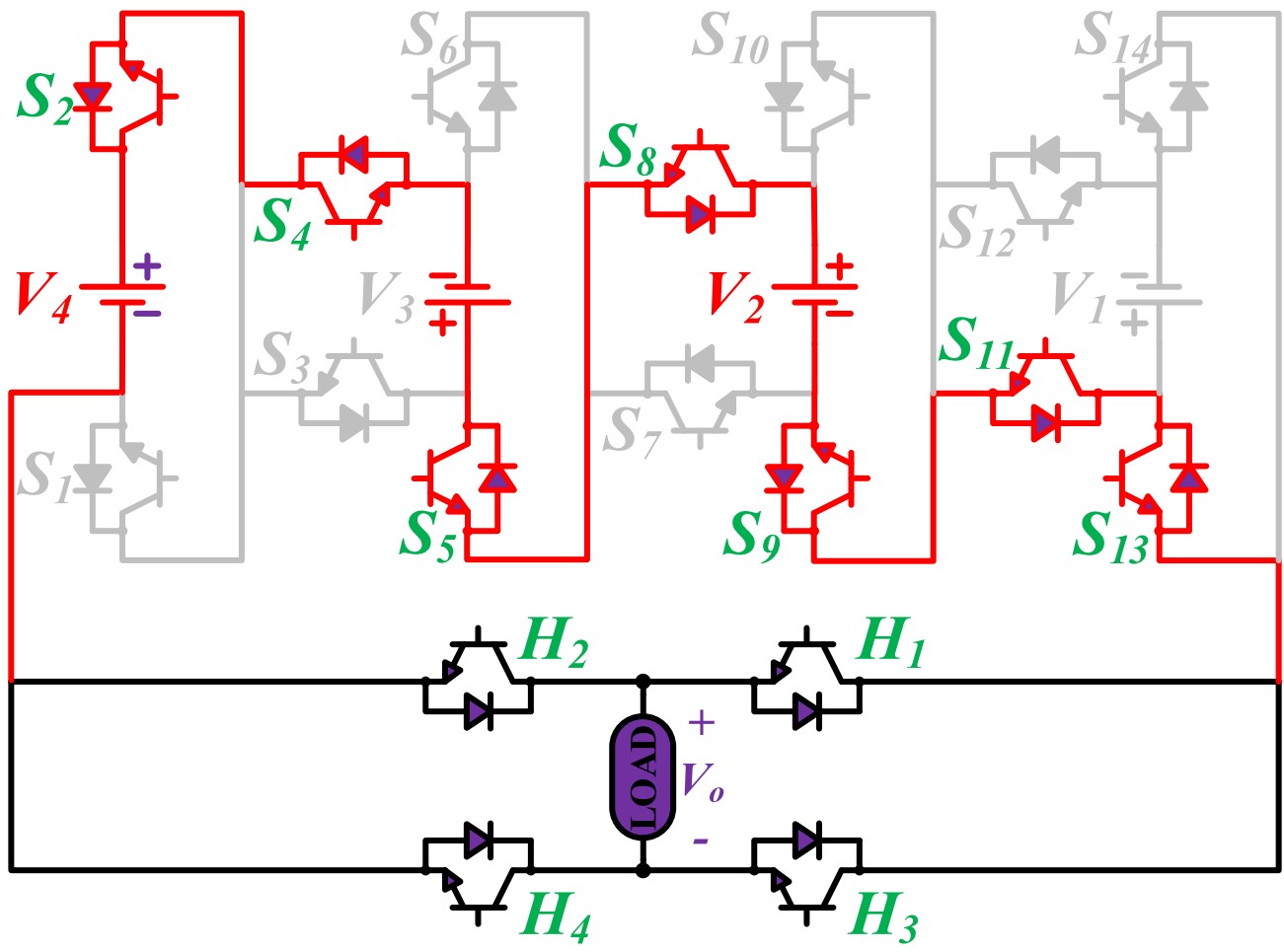

**Fig 11. Switching states for voltage magnitude level 33.**

The Eqs (10)–(15) are used for the calculation of practical efficiency of the proposed 81-level MWMLI topology at various loading conditions. The data of real MOSFETs used in the hardware prototype is utilized to calculate the efficiency for various loads at unity PF and 0.8PF lagging. The resulting plot for efficiency versus output power is shown in Fig 26.

## 4 Comparative analysis

The proposed 81-level MWMLI is compared with other topologies from the literature for various essential parameters such as the number of switches $N_S$, number of isolated gate drivers $N_{GD}$, number of DC voltage sources $N_{DC}$, number of diodes $N_D$, number of capacitors $N_C$, components count per level factor $F_{CPL}$, percent THD and TSV. For a meaningful comparison, the design equations from those papers are used to calculate the parameters for 81-level output voltage. The comparison of all the parameters mentioned above is shown in Table 3 and the graphical comparison for various parameters is shown in Figs 27–29.

## 5 Discussion

The proposed 81-level MWMLI is simulated on MATLAB/Simulink and verified experimentally. The output voltage waveform shows that the proposed MWMLI configuration generates

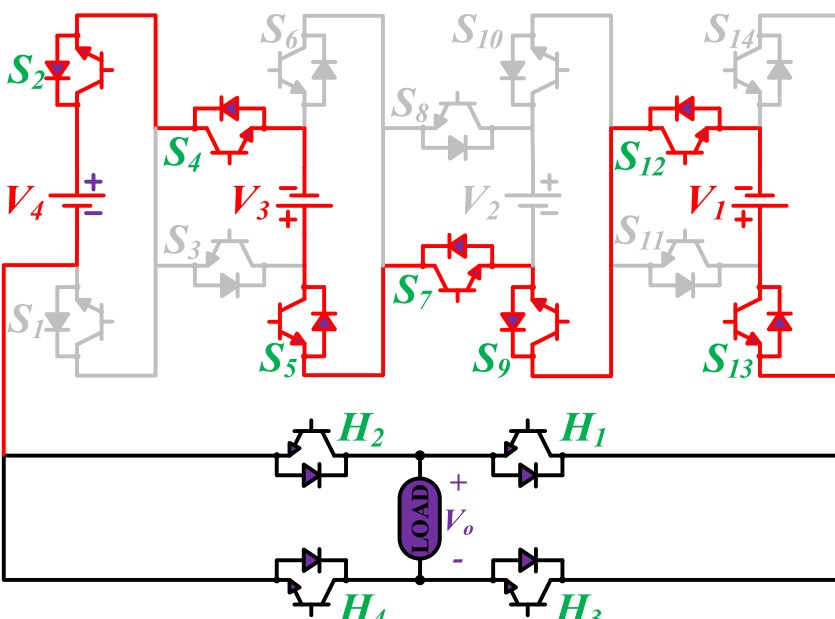

**Fig 12. Switching states for voltage magnitude level 37.**

an approaching sinusoidal voltage without using any output filter. The FFT and THD analysis depict the output voltage THD of 1.04%, which is within limits specified by IEEE 519 standard for a medium voltage system. The efficiency plot shows that the efficiency of the proposed MWMLI for resistive loads is above 90%, whereas for 0.8PF loads is slightly less than 90%. The efficiency plot also shows that the efficiency starts to decrease slightly by increasing the output power. This effect is justified as the conduction loss depends on the current square. The

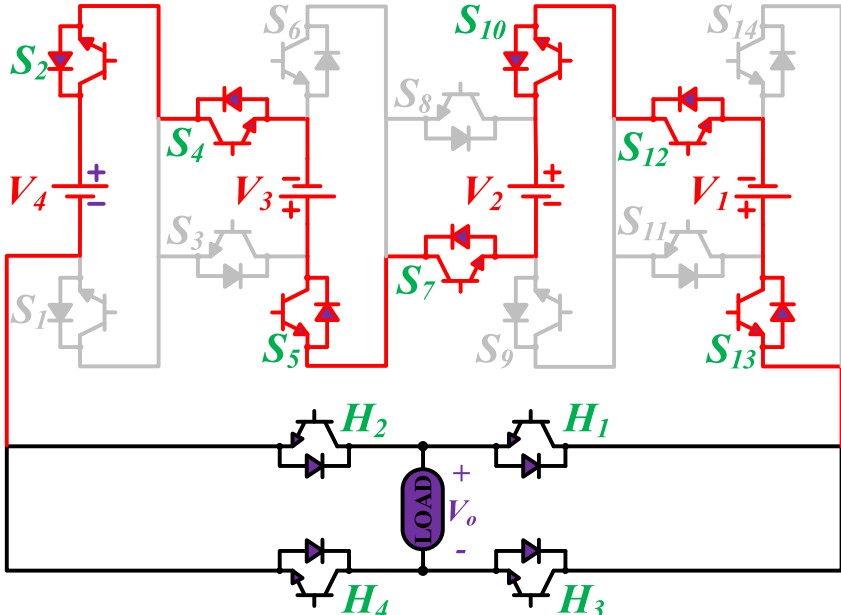

**Fig 13. Switching states for voltage magnitude level 40.**

**Table 1. Switching states for the 81-level MWMLI.**

| $S_1$ | $S_3$ | $S_5$ | $S_7$ | $S_9$ | $S_{11}$ | $S_{13}$ | $H_1$ | $H_3$ | $V_O$ | $S_1$ | $S_3$ | $S_5$ | $S_7$ | $S_9$ | $S_{11}$ | $S_{13}$ | $H_1$ | $H_3$ | $V_O$ |
|---|---|---|---|---|---|---|---|---|---|---|---|---|---|---|---|---|---|---|---|
| 1 | 1 | 1 | 1 | 1 | 1 | 1 | 1 | 1 | *0* | 1 | 1 | 1 | 1 | 1 | 1 | 1 | 1 | 1 | *0* |
| 1 | 1 | 1 | 1 | 1 | 0 | 1 | 1 | 0 | *V* | 1 | 1 | 1 | 1 | 1 | 0 | 1 | 0 | 1 | *-V* |
| 1 | 1 | 1 | 1 | 0 | 1 | 0 | 1 | 0 | *2V* | 1 | 1 | 1 | 1 | 0 | 1 | 0 | 0 | 1 | *-2V* |
| 1 | 1 | 1 | 1 | 0 | 0 | 0 | 1 | 0 | *3V* | 1 | 1 | 1 | 1 | 0 | 0 | 0 | 0 | 1 | *-3V* |
| 1 | 1 | 1 | 1 | 0 | 0 | 1 | 1 | 0 | *4V* | 1 | 1 | 1 | 1 | 0 | 0 | 1 | 0 | 1 | *-4V* |
| 1 | 0 | 1 | 0 | 1 | 1 | 0 | 1 | 0 | *5V* | 1 | 0 | 1 | 0 | 1 | 1 | 0 | 0 | 1 | *-5V* |
| 1 | 0 | 1 | 0 | 1 | 1 | 1 | 1 | 0 | *6V* | 1 | 0 | 1 | 0 | 1 | 1 | 1 | 0 | 1 | *-6V* |
| 1 | 0 | 1 | 0 | 1 | 0 | 1 | 1 | 0 | *7V* | 1 | 0 | 1 | 0 | 1 | 0 | 1 | 0 | 1 | *-7V* |
| 1 | 0 | 1 | 1 | 1 | 1 | 0 | 1 | 0 | *8V* | 1 | 0 | 1 | 1 | 1 | 1 | 0 | 0 | 1 | *-8V* |
| 1 | 0 | 1 | 1 | 1 | 1 | 1 | 1 | 0 | *9V* | 1 | 0 | 1 | 1 | 1 | 1 | 1 | 0 | 1 | *-9V* |
| 1 | 0 | 1 | 1 | 1 | 0 | 1 | 1 | 0 | *10V* | 1 | 0 | 1 | 1 | 1 | 0 | 1 | 0 | 1 | *-10V* |
| 1 | 0 | 1 | 1 | 0 | 1 | 0 | 1 | 0 | *11V* | 1 | 0 | 1 | 1 | 0 | 1 | 0 | 0 | 1 | *-11V* |
| 1 | 0 | 1 | 1 | 0 | 1 | 1 | 1 | 0 | *12V* | 1 | 0 | 1 | 1 | 0 | 1 | 1 | 0 | 1 | *-12V* |
| 1 | 0 | 1 | 1 | 0 | 0 | 1 | 1 | 0 | *13V* | 1 | 0 | 1 | 1 | 0 | 0 | 1 | 0 | 1 | *-13V* |
| 0 | 1 | 0 | 0 | 1 | 1 | 0 | 1 | 0 | *14V* | 0 | 1 | 0 | 0 | 1 | 1 | 0 | 0 | 1 | *-14V* |
| 0 | 1 | 0 | 0 | 1 | 1 | 1 | 1 | 0 | *15V* | 0 | 1 | 0 | 0 | 1 | 1 | 1 | 0 | 1 | *-15V* |
| 0 | 1 | 0 | 0 | 1 | 0 | 1 | 1 | 0 | *16V* | 0 | 1 | 0 | 0 | 1 | 0 | 1 | 0 | 1 | *-16V* |
| 0 | 1 | 0 | 1 | 1 | 1 | 0 | 1 | 0 | *17V* | 0 | 1 | 0 | 1 | 1 | 1 | 0 | 0 | 1 | *-17V* |
| 0 | 1 | 0 | 1 | 1 | 1 | 1 | 1 | 0 | *18V* | 0 | 1 | 0 | 1 | 1 | 1 | 1 | 0 | 1 | *-18V* |
| 0 | 1 | 0 | 1 | 1 | 0 | 1 | 1 | 0 | *19V* | 0 | 1 | 0 | 1 | 1 | 0 | 1 | 0 | 1 | *-19V* |
| 0 | 1 | 0 | 1 | 0 | 1 | 0 | 1 | 0 | *20V* | 0 | 1 | 0 | 1 | 0 | 1 | 0 | 0 | 1 | *-20V* |
| 0 | 1 | 0 | 1 | 0 | 1 | 1 | 1 | 0 | *21V* | 0 | 1 | 0 | 1 | 0 | 1 | 1 | 0 | 1 | *-21V* |
| 0 | 1 | 0 | 1 | 0 | 0 | 1 | 1 | 0 | *22V* | 0 | 1 | 0 | 1 | 0 | 0 | 1 | 0 | 1 | *-22V* |
| 0 | 1 | 1 | 0 | 1 | 1 | 0 | 1 | 0 | *23V* | 0 | 1 | 1 | 0 | 1 | 1 | 0 | 0 | 1 | *-23V* |
| 0 | 1 | 1 | 0 | 1 | 1 | 1 | 1 | 0 | *24V* | 0 | 1 | 1 | 0 | 1 | 1 | 1 | 0 | 1 | *-24V* |
| 0 | 1 | 1 | 0 | 1 | 0 | 1 | 1 | 0 | *25V* | 0 | 1 | 1 | 0 | 1 | 0 | 1 | 0 | 1 | *-25V* |
| 0 | 1 | 1 | 1 | 1 | 1 | 0 | 1 | 0 | *26V* | 0 | 1 | 1 | 1 | 1 | 1 | 0 | 0 | 1 | *-26V* |
| 0 | 1 | 1 | 1 | 1 | 1 | 1 | 1 | 0 | *27V* | 0 | 1 | 1 | 1 | 1 | 1 | 1 | 0 | 1 | *-27V* |
| 0 | 1 | 1 | 1 | 1 | 0 | 1 | 1 | 0 | *28V* | 0 | 1 | 1 | 1 | 1 | 0 | 1 | 0 | 1 | *-28V* |
| 0 | 1 | 1 | 1 | 0 | 1 | 0 | 1 | 0 | *29V* | 0 | 1 | 1 | 1 | 0 | 1 | 0 | 0 | 1 | *-29V* |
| 0 | 1 | 1 | 1 | 0 | 1 | 1 | 1 | 0 | *30V* | 0 | 1 | 1 | 1 | 0 | 1 | 1 | 0 | 1 | *-30V* |
| 0 | 1 | 1 | 1 | 0 | 0 | 1 | 1 | 0 | *31V* | 0 | 1 | 1 | 1 | 0 | 0 | 1 | 0 | 1 | *-31V* |
| 0 | 0 | 1 | 0 | 1 | 1 | 0 | 1 | 0 | *32V* | 0 | 0 | 1 | 0 | 1 | 1 | 0 | 0 | 1 | *-32V* |
| 0 | 0 | 1 | 0 | 1 | 1 | 1 | 1 | 0 | *33V* | 0 | 0 | 1 | 0 | 1 | 1 | 1 | 0 | 1 | *-33V* |
| 0 | 0 | 1 | 0 | 1 | 0 | 1 | 1 | 0 | *34V* | 0 | 0 | 1 | 0 | 1 | 0 | 1 | 0 | 1 | *-34V* |
| 0 | 0 | 1 | 1 | 1 | 1 | 0 | 1 | 0 | *35V* | 0 | 0 | 1 | 1 | 1 | 1 | 0 | 0 | 1 | *-35V* |
| 0 | 0 | 1 | 1 | 1 | 1 | 1 | 1 | 0 | *36V* | 0 | 0 | 1 | 1 | 1 | 1 | 1 | 0 | 1 | *-36V* |
| 0 | 0 | 1 | 1 | 1 | 0 | 1 | 1 | 0 | *37V* | 0 | 0 | 1 | 1 | 1 | 0 | 1 | 0 | 1 | *-37V* |
| 0 | 0 | 1 | 1 | 0 | 1 | 0 | 1 | 0 | *38V* | 0 | 0 | 1 | 1 | 0 | 1 | 0 | 0 | 1 | *-38V* |
| 0 | 0 | 1 | 1 | 0 | 1 | 1 | 1 | 0 | *39V* | 0 | 0 | 1 | 1 | 0 | 1 | 1 | 0 | 1 | *-39V* |
| 0 | 0 | 1 | 1 | 0 | 0 | 1 | 1 | 0 | *40V* | 0 | 0 | 1 | 1 | 0 | 0 | 1 | 0 | 1 | *-40V* |

*V* is the lowest DC voltage in the circuit (i.e. $V_1$).

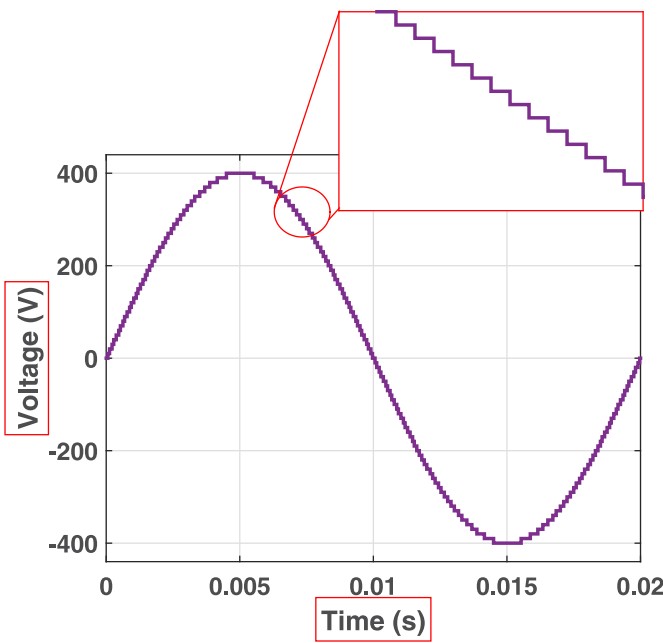

**Fig 14. Output voltage of 81-level MWMLI with zoomed-in levels.**

drawback of the proposed topology is that it is slightly less efficient than the other available topologies in the literature. However, the output power quality is improved due to the increased output voltage levels compared to those topologies. As seen in the comparative analysis section, the proposed MWMLI topology uses fewer hardware resources than other topologies to generate the same number of voltage levels.

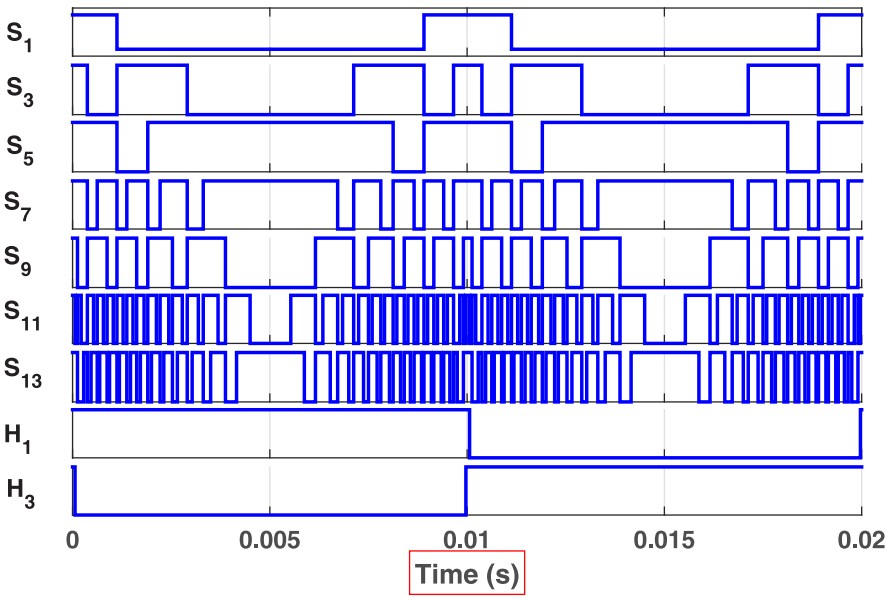

**Fig 15. PWM waveform for the 81-level proposed MWMLI.**

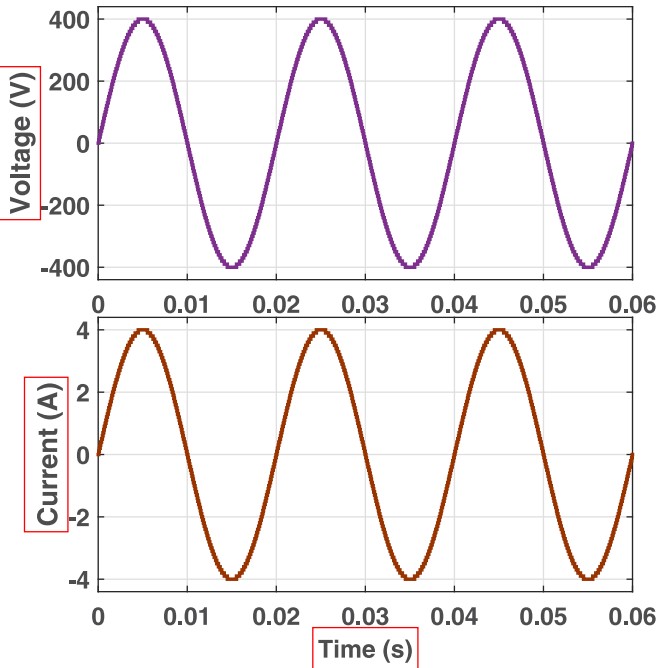

**Fig 16. Simulation waveforms (voltage and current) at the output of 81-level MWMLI for R-load.**

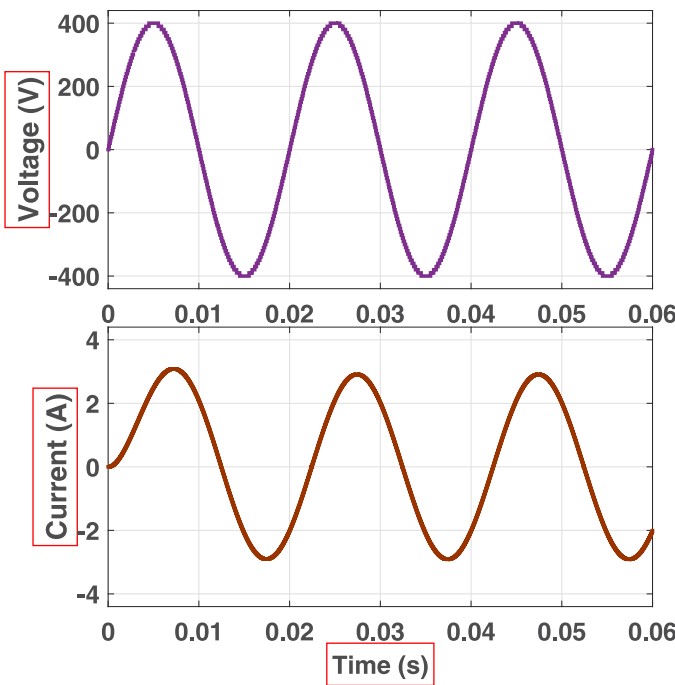

**Fig 17. Simulation waveforms (voltage and current) at the output of 81-level MWMLI for RL-load.**

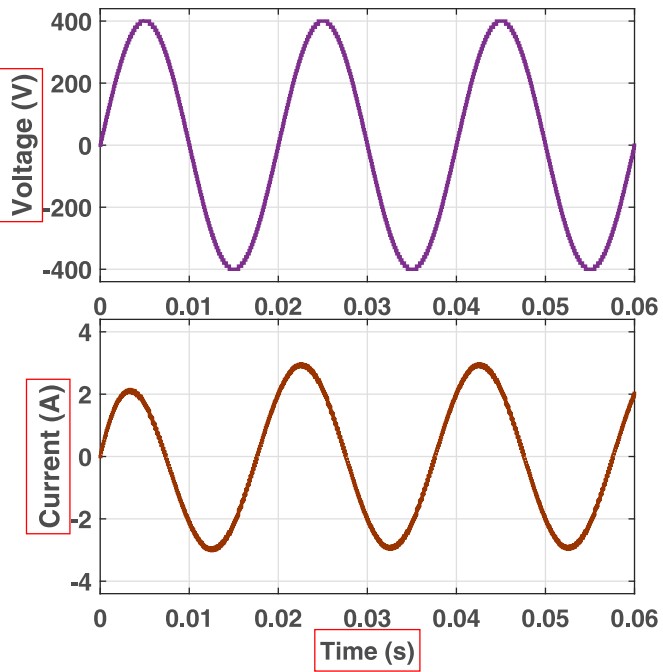

**Fig 18. Simulation waveforms (voltage and current) at the output of 81-level MWMLI for RC-load.**

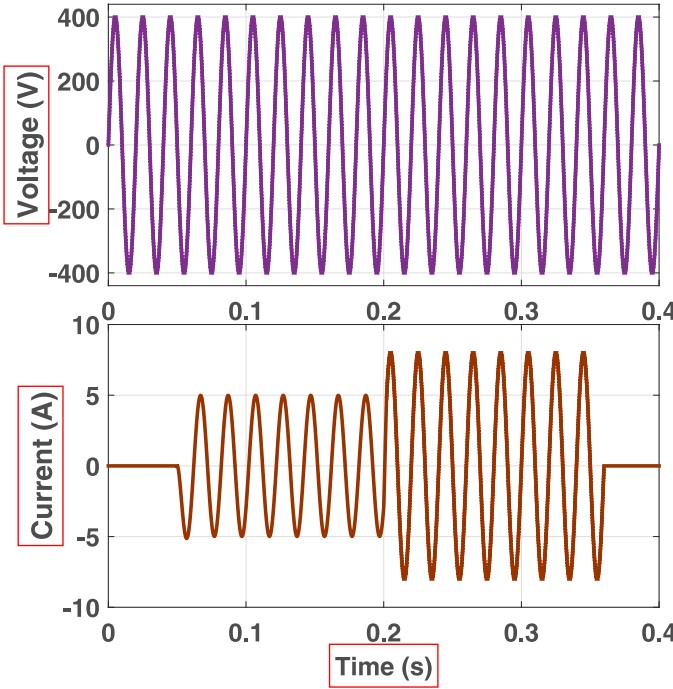

**Fig 19. Effect of dynamic load switching on 81-level MWMLI: No-load → 1 kVA @0.8 PF lagging → 1.6 kW → no-load.**

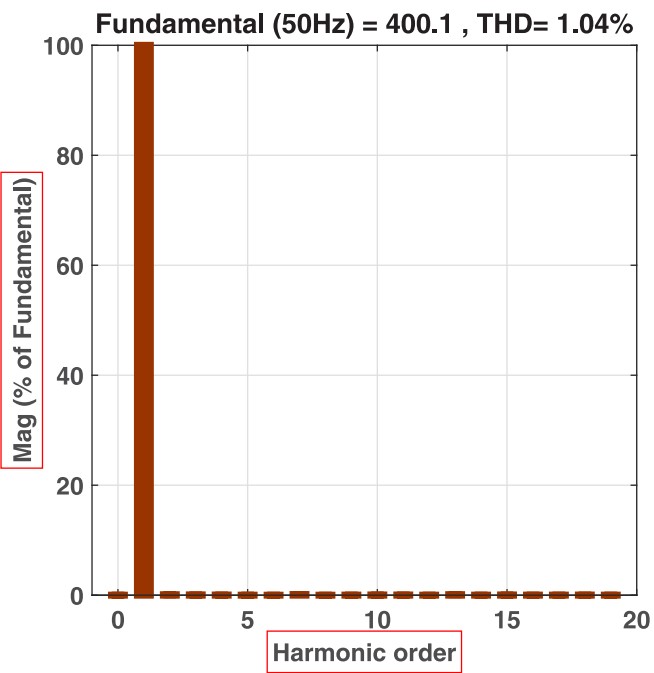

**Fig 20. FFT and THD analysis of the output voltage for 81-level MWMLI.**

**Table 2. Parameters and main components used for hardware prototype development.**

| Components/Parameters | Quantity/Value |
|---|---|
| Working Output Voltage | 400 V |
| Operating Frequency | 50 Hz |
| Power Rating of Prototype | 200 W |
| Voltages of DC Power Sources | 270 V, 90 V, 30 V, 10 V |
| IRFP450 | 6 |
| R6015ENX | 8 |
| KNF6450A | 4 |
| TLP250 | 16 |
| IR2103 | 2 |

## 6 Conclusion

In this paper, a new 81-level MWMLI is presented. The proposed inverter has a reduced components count and uses only 18 unidirectional switches to generate an 81-level output voltage. The 81-level MWMLI is simulated in MATLAB/Simulink, and the results are validated using a hardware prototype. The proposed 81-level inverter is analyzed for various important parameters such as number of switches, number of levels, number of isolated gate drivers, THD and TSV. After performing a thorough analysis of the proposed MLI topology, a comparative analysis is carried out with the other available inverter topologies that confirm its novelty. It was observed that the MWMLI topology is exceptionally suitable in terms of cost and effectiveness. It provides the best resolution in terms of the number of levels at the output, and the resolution

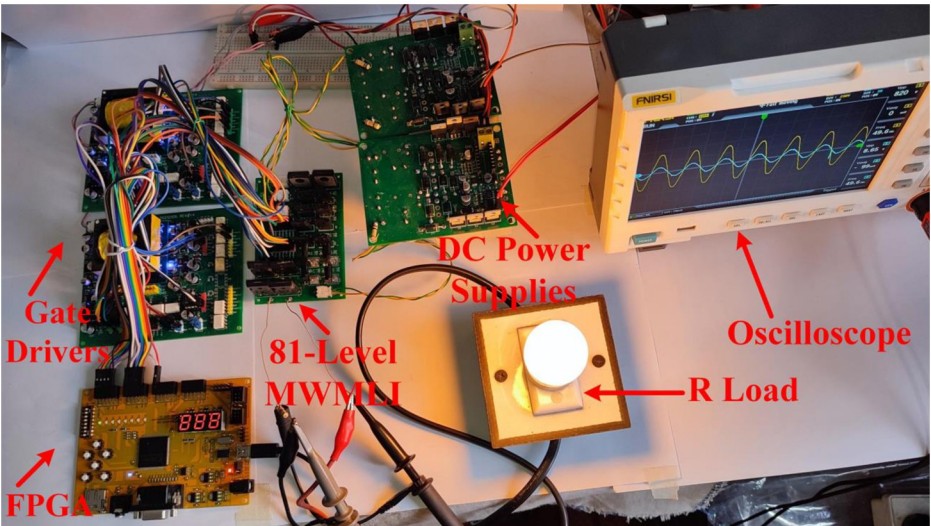

**Fig 21. Hardware prototype for the 81-level MWMLI.**

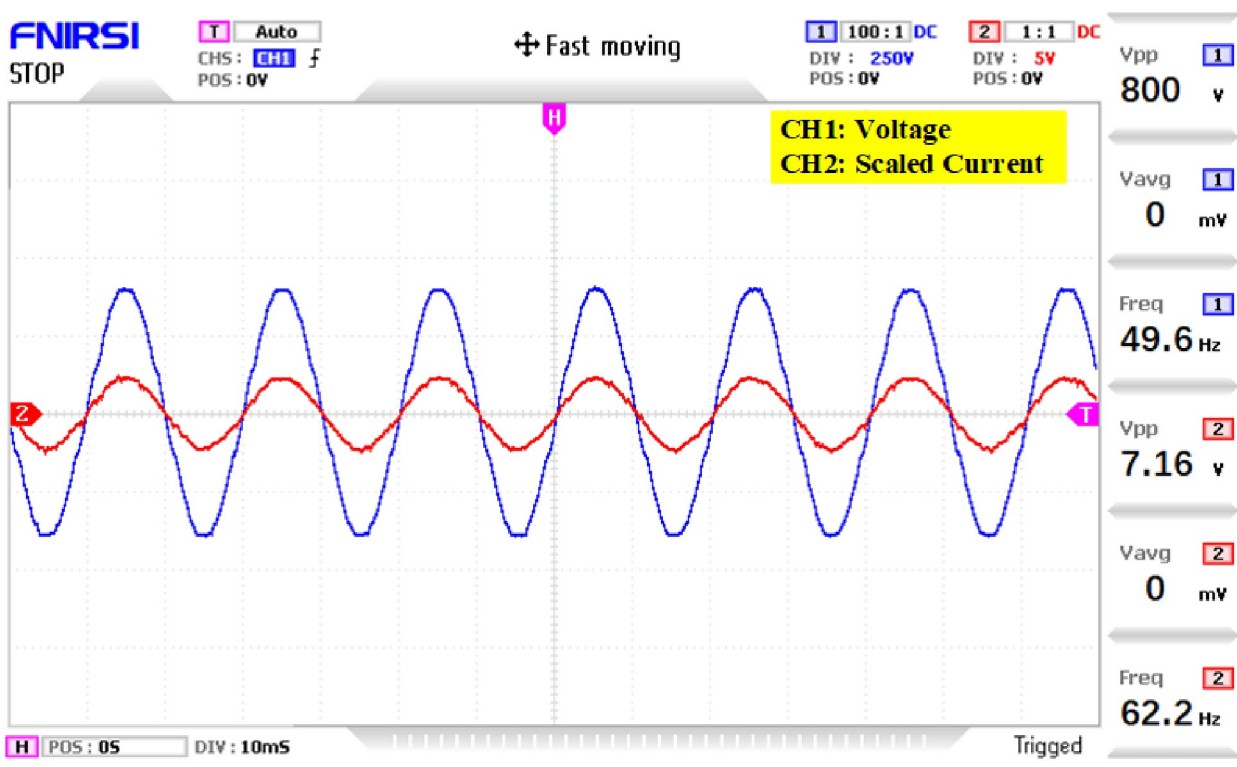

**Fig 22. Experimental waveforms (voltage and current) at the output of 81-level MWMLI for resistive load.**

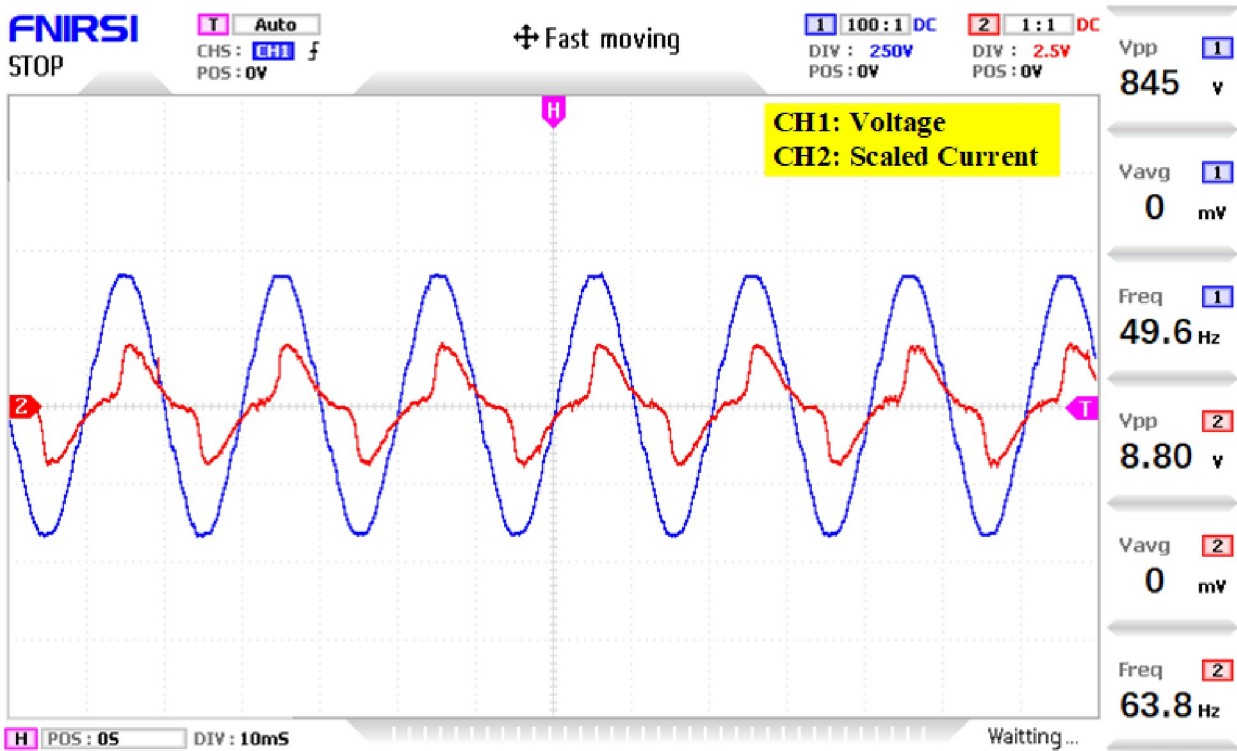

**Fig 23. Experimental waveforms (voltage and current) at the output of 81-level MWMLI for non-linear lagging load.**

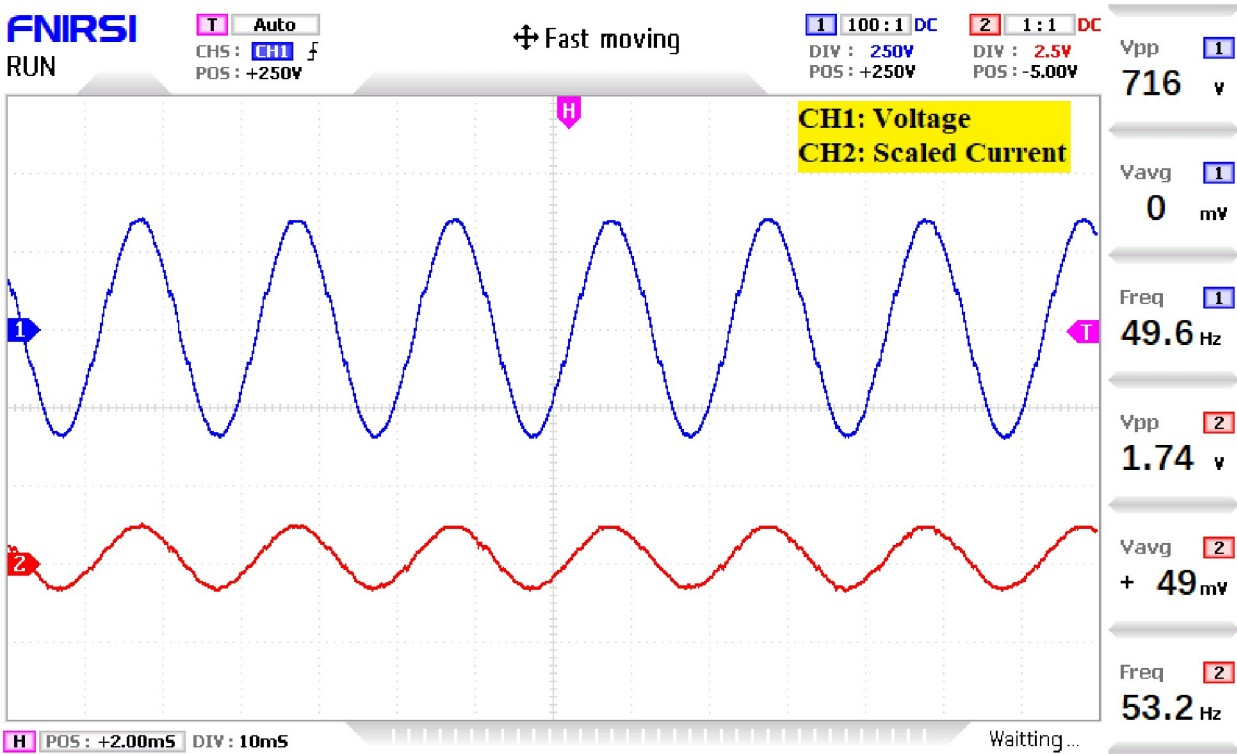

**Fig 24. Experimental waveforms (voltage and current) at the output of 81-level MWMLI for MI = 0.88 at unity PF load.**

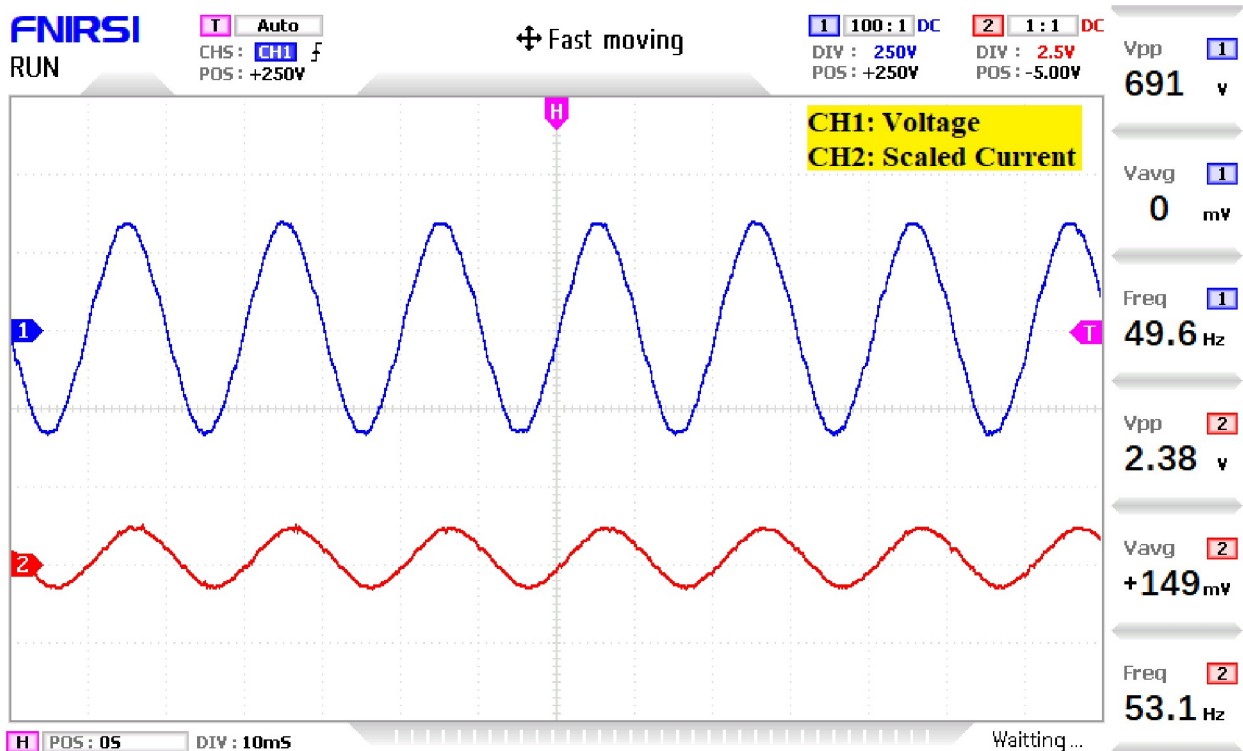

**Fig 25. Experimental waveforms (voltage and current) at the output of 81-level MWMLI for MI = 0.85 at 0.9PF lagging load.**

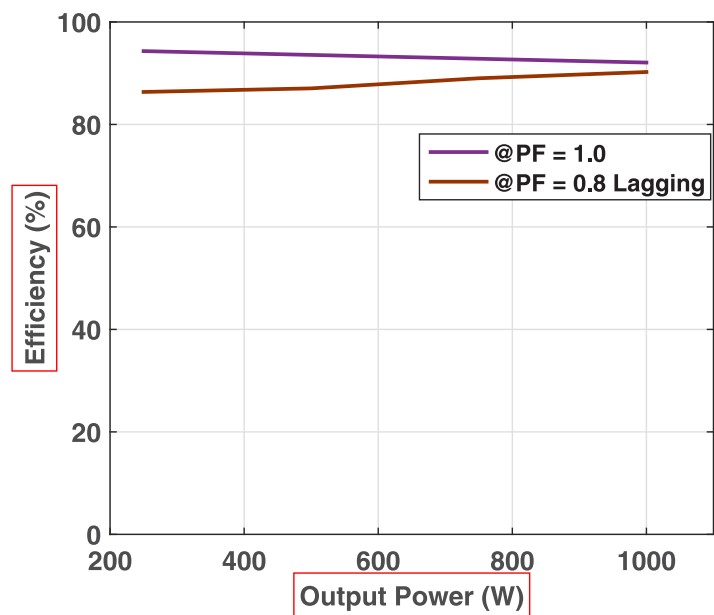

**Fig 26. Experimental efficiency plot for various resistive and inductive loads.**

**Table 3. Comparison between various MLI topologies.**

| Studies / Parameters | [5] | [19] | [27] | [28] | [29] | [30] | [31] | Proposed |
|---|---|---|---|---|---|---|---|---|
| $N_L$ | 81 | 81 | 81 | 81 | 81 | 81 | 81 | 81 |
| $N_S$ | 25 | 31 | 20 | 44 | 31 | 13 | 13 | 18 |
| $N_{DC}$ | 20 | 15 | 8 | 2 | 16 | 7 | 7 | 4 |
| $N_{GD}$ | 25 | 31 | 16 | 25 | 31 | 13 | 13 | 8 |
| $N_D$ | 80 | not required | not required | 44 | 25 | 13 | 35 | not required |
| $N_C$ | 20 | not required | not required | 20 | not required | not required | not required | not required |
| $F_{CPL}$ | 2.1 | 0.95 | 0.54 | 1.67 | 1.27 | 0.57 | 0.84 | 0.37 |
| %THD | - | - | 0.94 | - | - | - | - | 1.04 |
| TSV (V) | - | $164V$ | $280V$ | - | $197V$ | - | - | $266V$ |

$V$ is the lowest DC voltage in the circuit (i.e, $V_1$).

can be controlled with the help of the PWM of the switches. Therefore, any number of odd voltage levels up to eighty-one can be obtained from the proposed configuration. Loss analysis is also carried out to model the conduction/switching losses and efficiency. The proposed inverter configuration can be used as a standalone system or integrated with the electric power grid with reduced harmonic content.

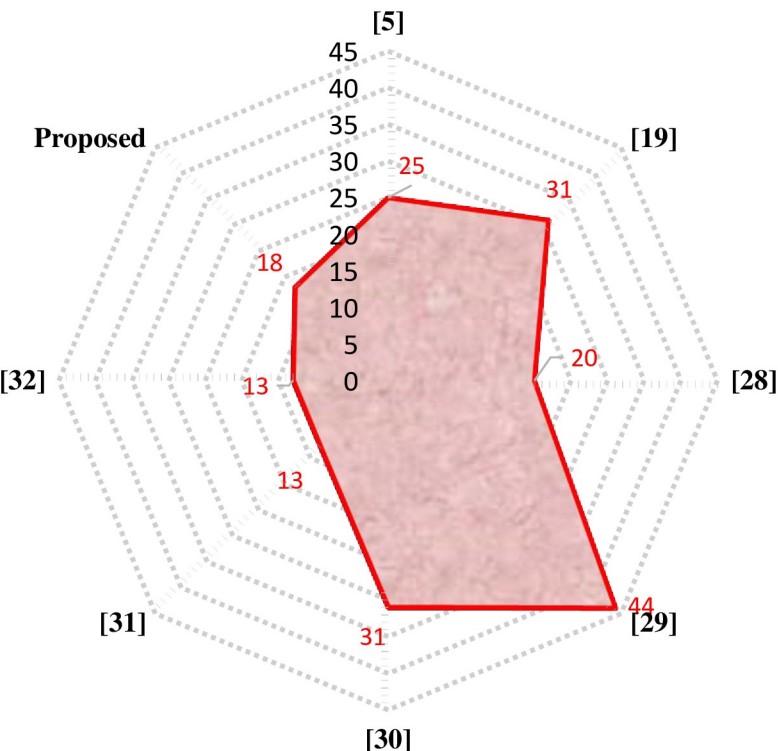

**Fig 27. Comparison of 81-level MWMLI with various topologies for the number of switches as a parameter.**

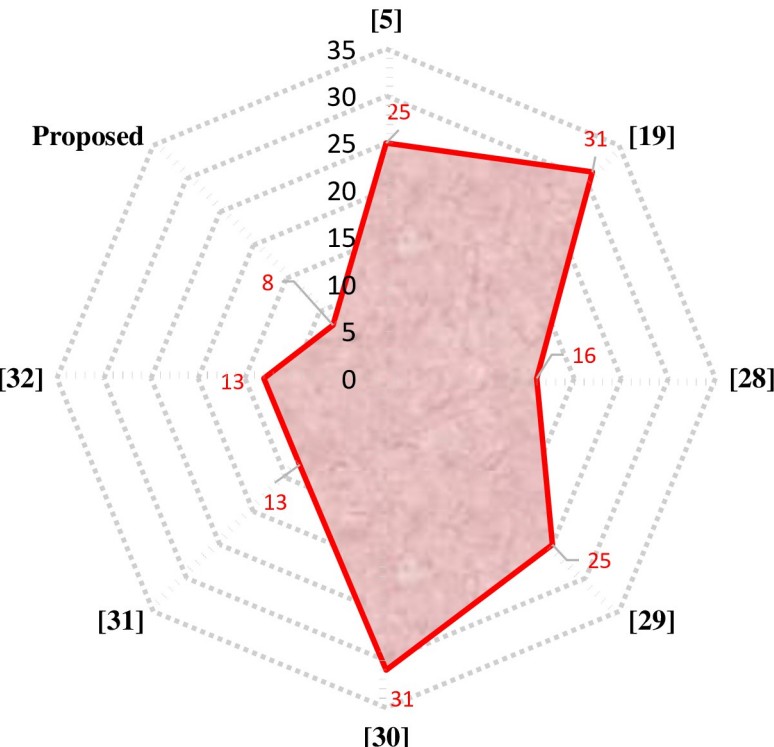

**Fig 28. Comparison of 81-level MWMLI with various topologies for the number of isolated gate drivers as a parameter.**

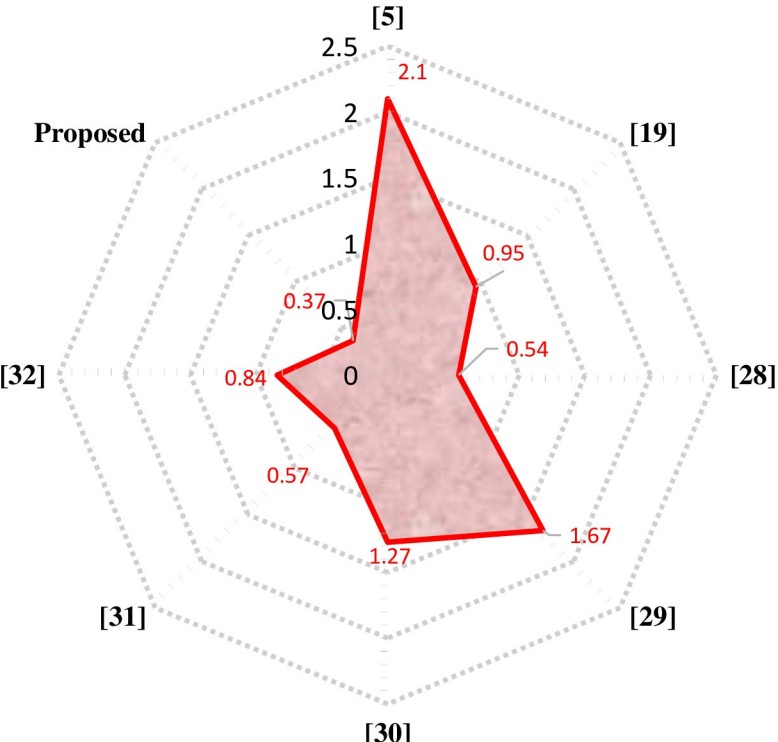

**Fig 29. Comparison of 81-level MWMLI with various topologies for the components per level factor as a parameter.**

## Supporting information

**S1 Fig. PV scheme.**
(EPS)

**S2 Fig. Gate driver circuit.**
(PDF)

## Author Contributions

**Conceptualization:** Sajjad Ahmed.

**Data curation:** Sajjad Ahmed.

**Formal analysis:** Sajjad Ahmed.

**Investigation:** Sajjad Ahmed.

**Methodology:** Sajjad Ahmed.

**Resources:** Sajjad Ahmed.

**Software:** Sajjad Ahmed.

**Supervision:** Muhammad Asghar Saqib, Syed Abdul Rahman Kashif.

**Validation:** Sajjad Ahmed.

**Visualization:** Sajjad Ahmed.

**Writing – original draft:** Sajjad Ahmed.

**Writing – review & editing:** Sajjad Ahmed, Muhammad Asghar Saqib, Syed Abdul Rahman Kashif.

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
