## [Decision Letter · Decision Letter 0]

24 Feb 2022

PONE-D-21-38322Modified W-Type Configuration for a Single-Phase Reduced Parts Count 81-Level InverterPLOS ONE

Dear Dr. Saqib,

Thank you for submitting your manuscript to PLOS ONE. After careful consideration, we feel that it has merit but does not fully meet PLOS ONE’s publication criteria as it currently stands. Therefore, we invite you to submit a revised version of the manuscript that addresses the points raised during the review process.

We look forward to receiving your revised manuscript.

Kind regards,

Yogendra Arya

Academic Editor

PLOS ONE

Journal Requirements:

3. Please ensure that you refer to Figures 3 to 12, 16 and 20 in your text as, if accepted, production will need this reference to link the reader to the figure.

Reviewers' comments:

Reviewer's Responses to Questions

**Comments to the Author**

1. Is the manuscript technically sound, and do the data support the conclusions?

Reviewer #1: Yes

Reviewer #2: Yes

Reviewer #3: No

2. Has the statistical analysis been performed appropriately and rigorously? 

Reviewer #1: N/A

Reviewer #2: Yes

Reviewer #3: No

3. Have the authors made all data underlying the findings in their manuscript fully available?

Reviewer #1: Yes

Reviewer #2: Yes

Reviewer #3: Yes

4. Is the manuscript presented in an intelligible fashion and written in standard English?

Reviewer #1: Yes

Reviewer #2: No

Reviewer #3: Yes

5. Review Comments to the Author

Reviewer #1: The general idea of the paper seems to be good. However, the paper organization is not acceptable, and there are several major technical challenges that should be effectively addressed.

Comments:

1. Need more experimental results by varying load, and modulation index.

2. Include the efficiency of the proposed inverter

3. The abstract has been briefly written and should be enriched by adding the main ideas and contributions.

4. There are some grammatical errors and typos that should be corrected before publication.

5. It is recommended to provide a nomenclature at the beginning of the paper to define all variables clearly.

6. The introduction has been vaguely written. My suggestion is to divide the introduction into three subsections: 1) motivation and incitement, 2) literature review and 3) contribution and paper organization.

7. The main contribution of the paper should be highlighted and emphasized. It would be great if the drawbacks and gaps of literature are clear and, particularly, how the proposed approach aims at filling these gaps.

8. Simulation Results should be enriched. More detailed scenarios should be added.

9. A separate section should be added for discussion of obtained results and main achievements.

10. The quality of all the figures is poor and should be improved.

11. Need to update the following papers and compare with proposed topology.

Reviewer #2: This paper presents a Modified W-Type Configuration for a Single-Phase Reduced Parts Count 81-Level Inverter; the reviewer has the following comments and questions for the authors to address:

1. Abstract needs to be rewritten to include some significant results

2. More results should be added at different loading conditions/input voltage

3. Add a photo of the experimental setup if available

4. Discussion on the limitation of the proposed MLI topology should added

5. The paper requires further English revision as it has many grammatical mistakes

6. A detailed comparison should be presented, with what has been presented in the literature ( use a table format with specific parameters such as the number of switches, no. of diodes, passive components….)

Reviewer #3: Dear Authors,

My suggestions to improve the quality of the manuscript are as follows:

1. Authors stated in the abstract, "The proposed topology offers very low total harmonic distortion of the output voltage that is within the benchmarks specified for grid integration". My 1st major concern is to developing a grid-integrated system.

If not able to implement experimentally, the simulation results of the same must be incorporated in the revised manuscript.

2. The second major concern is the availability of DC sources, cost of DC sources, What if the DC sources are replaced by the PV sources for the grid integration. Elaborate the scheme.

3. Third concern is about the blocking voltage of the switches. How, the selection of the power switches are made as the DC sources are diverse.

4. The comparison made is not sufficient. Please compare with the papers as similar to following:

Experimental Verification of a New Scheme of MLI Based on Modified T-Type Inverter and Switched-Diode Cell with Lower Number of Circuit Devices

A New and Generalized MLI with Overall Lesser Power Electronic Devices

A new and modular back-to-back connected t-type inverter for minimum number of power devices, tsv, and cost factor

Optimum structure based Multi-level Inverter with doubling circuit configuration

Reduction of power electronic devices in a single-phase generalized multilevel inverter

Analysis and implementation of a generalised switched‐capacitor multi‐level inverter having the lower total standing voltage

A new and generalized structure of MLI topology with half-bridge cell with minimum number of power electronic devices

N-Level Cascade Multilevel Converter with optimum number of switches

Reduction in controlled power switches for a single-phase novel multilevel inverter

5. Fourth major concern is about the loss analysis and experimental results. The experimental results of the dynamic condition are missing. No data for the the loss analysis (conduction losses and switching losses) and efficiency is provided. Justify.

6. Include the detailed circuit diagram, operation for the driver circuit and the delay circuit.

6. PLOS authors have the option to publish the peer review history of their article (what does this mean?). If published, this will include your full peer review and any attached files.

Reviewer #1: No

Reviewer #2: No

Reviewer #3: No

---

## [Author Response · Author response to Decision Letter 0]

9 Mar 2022

All the comments of editor and reviewers are responded in the "Response to Reviewers" file.

---

## [Decision Letter · Decision Letter 1]

4 May 2022

PONE-D-21-38322R1Modified W-type configuration for a single-phase reduced parts count 81-level inverterPLOS ONE

Dear Dr. Saqib,

Thank you for submitting your manuscript to PLOS ONE. After careful consideration, we feel that it has merit but does not fully meet PLOS ONE’s publication criteria as it currently stands. Therefore, we invite you to submit a revised version of the manuscript that addresses the points raised during the review process.

We look forward to receiving your revised manuscript.

Kind regards,

Yogendra Arya

Academic Editor

PLOS ONE

Journal Requirements:

Additional Editor Comments (if provided):

One reviewer has asked the authors for thoroughly proofreading of the paper before final acceptance.

Hence, authors are advised to do the same with sincerity and effectively.

Reviewers' comments:

Reviewer's Responses to Questions

**Comments to the Author**

1. If the authors have adequately addressed your comments raised in a previous round of review and you feel that this manuscript is now acceptable for publication, you may indicate that here to bypass the “Comments to the Author” section, enter your conflict of interest statement in the “Confidential to Editor” section, and submit your "Accept" recommendation.

Reviewer #1: All comments have been addressed

Reviewer #2: All comments have been addressed

Reviewer #3: All comments have been addressed

2. Is the manuscript technically sound, and do the data support the conclusions?

Reviewer #1: Yes

Reviewer #2: Yes

Reviewer #3: Yes

3. Has the statistical analysis been performed appropriately and rigorously? 

Reviewer #1: Yes

Reviewer #2: Yes

Reviewer #3: Yes

4. Have the authors made all data underlying the findings in their manuscript fully available?

Reviewer #1: Yes

Reviewer #2: Yes

Reviewer #3: Yes

5. Is the manuscript presented in an intelligible fashion and written in standard English?

Reviewer #1: No

Reviewer #2: Yes

Reviewer #3: Yes

6. Review Comments to the Author

Reviewer #1: Authors may include more details about the applications because this single phase system will not suitable or high power applications. All my comments are addressed no more questions.

Reviewer #2: most of the comments have been addressed, just need to do proofreading before acceptance, most of the comments have been addressed, just need to do proofreading before acceptance

Reviewer #3: Dear Authors,

The paper now can be accepted in its current form. The comment have been addressed.

7. PLOS authors have the option to publish the peer review history of their article (what does this mean?). If published, this will include your full peer review and any attached files.

Reviewer #1: No

Reviewer #2: No

Reviewer #3: No

---

## [Author Response · Author response to Decision Letter 1]

6 May 2022

The answers to the reviewers' and editor's questions are addressed in "Response to Reviewers.docx".

---

## [Decision Letter · Decision Letter 2]

27 May 2022

Modified W-type configuration for a single-phase reduced parts count 81-level inverter

PONE-D-21-38322R2

Dear Dr. Saqib,

We’re pleased to inform you that your manuscript has been judged scientifically suitable for publication and will be formally accepted for publication once it meets all outstanding technical requirements.

Kind regards,

Yogendra Arya

Academic Editor

PLOS ONE

Additional Editor Comments (optional):

The paper can be accepted.

Reviewers' comments:

Reviewer's Responses to Questions

**Comments to the Author**

1. If the authors have adequately addressed your comments raised in a previous round of review and you feel that this manuscript is now acceptable for publication, you may indicate that here to bypass the “Comments to the Author” section, enter your conflict of interest statement in the “Confidential to Editor” section, and submit your "Accept" recommendation.

Reviewer #1: All comments have been addressed

Reviewer #2: All comments have been addressed

2. Is the manuscript technically sound, and do the data support the conclusions?

Reviewer #1: Yes

Reviewer #2: Yes

3. Has the statistical analysis been performed appropriately and rigorously? 

Reviewer #1: Yes

Reviewer #2: Yes

4. Have the authors made all data underlying the findings in their manuscript fully available?

Reviewer #1: Yes

Reviewer #2: Yes

5. Is the manuscript presented in an intelligible fashion and written in standard English?

Reviewer #1: Yes

Reviewer #2: Yes

6. Review Comments to the Author

Reviewer #1: No more comments and author presented the topology in nice way. The author response is satisfactory.

Reviewer #2: All the comments have been addressed.

7. PLOS authors have the option to publish the peer review history of their article (what does this mean?). If published, this will include your full peer review and any attached files.

Reviewer #1: No

Reviewer #2: No

---

## [Editor Report · Acceptance letter]

1 Jun 2022

PONE-D-21-38322R2 

Modified W-type configuration for a single-phase reduced parts count 81-level inverter 

Dear Dr. Saqib:

I'm pleased to inform you that your manuscript has been deemed suitable for publication in PLOS ONE. Congratulations! Your manuscript is now with our production department. 

Kind regards, 

on behalf of

Dr. Yogendra Arya 

Academic Editor

PLOS ONE